# Memory Limited, Streaming PCA

**Ioannis Mitliagkas**
Dept. of Electrical and Computer Engineering
The University of Texas at Austin
ioannis@utexas.edu

**Constantine Caramanis**
Dept. of Electrical and Computer Engineering
The University of Texas at Austin
constantine@utexas.edu

**Prateek Jain**
Microsoft Research
Bangalore, India
prajain@microsoft.com

## Abstract

We consider streaming, one-pass principal component analysis (PCA), in the high-dimensional regime, with limited memory. Here, $p$-dimensional samples are presented sequentially, and the goal is to produce the $k$-dimensional subspace that best approximates these points. Standard algorithms require $O(p^2)$ memory; meanwhile no algorithm can do better than $O(kp)$ memory, since this is what the output itself requires. Memory (or storage) complexity is most meaningful when understood in the context of computational and sample complexity. Sample complexity for high-dimensional PCA is typically studied in the setting of the *spiked covariance model*, where $p$-dimensional points are generated from a population covariance equal to the identity (white noise) plus a low-dimensional perturbation (the spike) which is the signal to be recovered. It is now well-understood that the spike can be recovered when the number of samples, $n$, scales proportionally with the dimension, $p$. Yet, all algorithms that provably achieve this, have memory complexity $O(p^2)$. Meanwhile, algorithms with memory-complexity $O(kp)$ do not have provable bounds on sample complexity comparable to $p$. We present an algorithm that achieves both: it uses $O(kp)$ memory (meaning storage of any kind) and is able to compute the $k$-dimensional spike with $O(p \log p)$ sample-complexity – the first algorithm of its kind. While our theoretical analysis focuses on the spiked covariance model, our simulations show that our algorithm is successful on much more general models for the data.

## 1 Introduction

Principal component analysis is a fundamental tool for dimensionality reduction, clustering, classification, and many more learning tasks. It is a basic preprocessing step for learning, recognition, and estimation procedures. The core computational element of PCA is performing a (partial) singular value decomposition, and much work over the last half century has focused on efficient algorithms (e.g., Golub & Van Loan (2012) and references therein) and hence on *computational complexity*.

The recent focus on understanding high-dimensional data, where the dimensionality of the data scales together with the number of available sample points, has led to an exploration of the *sample complexity* of covariance estimation. This direction was largely influenced by Johnstone's *spiked covariance model*, where data samples are drawn from a distribution whose (population) covariance is a low-rank perturbation of the identity matrix Johnstone (2001). Work initiated there, and also work done in Vershynin (2010a) (and references therein) has explored the power of batch PCA in the $p$-dimensional setting with sub-Gaussian noise, and demonstrated that the singular value decom-

position (SVD) of the empirical covariance matrix succeeds in recovering the principal components (extreme eigenvectors of the population covariance) with high probability, given $n = O(p)$ samples.

This paper brings the focus on another critical quantity: memory/storage. The only currently available algorithms with provable sample complexity guarantees either store all $n = O(p)$ samples (note that for more than a single pass over the data, the samples must all be stored) or explicitly form or approximate the empirical $p \times p$ (typically dense) covariance matrix. All cases require as much as $O(p^2)$ storage for exact recovery. In certain high-dimensional applications, where data points are high resolution photographs, biometrics, video, etc., $p$ often is of the order of $10^{10} - 10^{12}$, making the need for $O(p^2)$ memory prohibitive. At many computing scales, manipulating vectors of length $O(p)$ is possible, when storage of $O(p^2)$ is not. A typical desktop may have 10-20 GB of RAM, but will not have more than a few TB of total storage. A modern smart-phone may have as much as a GB of RAM, but has a few GB, not TB, of storage. In distributed storage systems, the scalability in storage comes at the heavy cost of communication.

In this light, we consider the *streaming data* setting, where the samples $\mathbf{x}_t \in \mathbb{R}^p$ are collected sequentially, and unless we store them, they are irretrievably gone.[1] On the *spiked covariance model* (and natural generalizations), we show that a simple algorithm requiring $O(kp)$ storage – the best possible – performs as well as batch algorithms (namely, SVD on the empirical covariance matrix), with sample complexity $O(p \log p)$. To the best of our knowledge, this is the only algorithm with both storage complexity and sample complexity guarantees.

We discuss connections to past work in detail in Section 2, introduce the model in Section 3, and present the solution to the rank 1 case, the rank $k$ case, and the perturbed-rank-$k$ case in Sections 4.1, 4.2 and 4.3, respectively. In Section 5 we provide experiments that not only confirm the theoretical results, but demonstrate that our algorithm works well outside the assumptions of our main theorems.

## 2   Related Work

Memory- and computation-efficient algorithms that operate on streaming data are plentiful in the literature and many seem to do well in practice. However, there is no algorithm that provably recovers the principal components in the same noise and sample-complexity regime as the batch PCA algorithm does *and* maintains a provably light memory footprint. Because of the practical relevance, there is renewed interest in this problem. The fact that it is an important unresolved issue has been pointed out in numerous places, e.g., Warmuth & Kuzmin (2008); Arora et al. (2012).

Online-PCA for *regret minimization* is considered in several papers, most recently in Warmuth & Kuzmin (2008). There the multiplicative weights approach is adapted to this problem, with experts corresponding to subspaces. The goal is to control the regret, improving on the natural follow-the-leader algorithm that performs batch-PCA at each step. However, the algorithm can require $O(p^2)$ memory, in order to store the multiplicative weights. A memory-light variant described in Arora et al. (2012) typically requires much less memory, but there are no guarantees for this, and moreover, for certain problem instances, its memory requirement is on the order of $p^2$.

Sub-sampling, dimensionality-reduction and sketching form another family of low-complexity and low-memory techniques, see, e.g., Clarkson & Woodruff (2009); Nadler (2008); Halko et al. (2011). These save on memory and computation by performing SVD on the resulting smaller matrix. The results in this line of work provide worst-case guarantees over the pool of data, and typically require a rapidly decaying spectrum, not required in our setting, to produce good bounds. More fundamentally, these approaches are not appropriate for data coming from a statistical model such as the spiked covariance model. It is clear that subsampling approaches, for instance, simply correspond to discarding most of the data, and for fundamental sample complexity reasons, cannot work. Sketching produces a similar effect: each column of the sketch is a random $(+/-)$ sum of the data points. If the data points are, e.g., independent Gaussian vectors, then so will each element of the sketch, and thus this approach again runs against fundamental sample complexity constraints. Indeed, it is straightforward to check that the guarantees presented in (Clarkson & Woodruff (2009); Halko et al. (2011)) are not strong enough to guarantee recovery of the spike. This is not because the results are weak; it is because they are geared towards worst-case bounds.

Algorithms focused on sequential SVD (e.g., Brand (2002, 2006), Comon & Golub (1990),Li (2004) and more recently Balzano et al. (2010); He et al. (2011)) seek to have the best subspace estimate at every time (i.e., each time a new data sample arrives) but without performing full-blown SVD at each step. While these algorithms indeed reduce both the computational and memory burden of batch-PCA, there are no rigorous guarantees on the quality of the principal components or on the *statistical performance* of these methods.

In a Bayesian mindset, some researchers have come up with expectation maximization approaches Roweis (1998); Tipping & Bishop (1999), that can be used in an incremental fashion. The finite sample behavior is not known.

Stochastic-approximation-based algorithms along the lines of Robbins & Monro (1951) are also quite popular, due to their low computational and memory complexity, and excellent performance. They go under a variety of names, including *Incremental PCA* (though the term *Incremental* has been used in the online setting as well Herbster & Warmuth (2001)), Hebbian learning, and stochastic power method Arora et al. (2012). The basic algorithms are some version of the following: upon receiving data point $\mathbf{x}_t$ at time $t$, update the estimate of the top $k$ principal components via:

$$U^{(t+1)} = \text{Proj}(U^{(t)} + \eta_t \mathbf{x}_t \mathbf{x}_t^\top U^{(t)}), \tag{1}$$

where $\text{Proj}(\cdot)$ denotes the "projection" that takes the SVD of the argument, and sets the top $k$ singular values to 1 and the rest to zero (see Arora et al. (2012) for discussion). While empirically these algorithms perform well, to the best of our knowledge - and efforts - there is no associated finite sample guarantee. The analytical challenge lies in the high variance at each step, which makes direct analysis difficult.

In summary, while much work has focused on memory-constrained PCA, there has as of yet been no work that simultaneously provides sample complexity guarantees competitive with batch algorithms, and also memory/storage complexity guarantees close to the minimal requirement of $O(kp)$ – the memory required to store only the output. We present an algorithm that provably does both.

## 3 Problem Formulation and Notation

We consider the streaming model: at each time step $t$, we receive a point $\mathbf{x}_t \in \mathbb{R}^p$. Any point that is not explicitly stored can never be revisited. Our goal is to compute the top $k$ principal components of the data: the $k$-dimensional subspace that offers the best squared-error estimate for the points. We assume a probabilistic generative model, from which the data is sampled at each step $t$. Specifically,

$$\mathbf{x}_t = A\mathbf{z}_t + \mathbf{w}_t, \tag{2}$$

where $A \in \mathbb{R}^{p \times k}$ is a fixed matrix, $\mathbf{z}_t \in \mathbb{R}^{k \times 1}$ is a multivariate normal random variable, i.e.,

$$\mathbf{z}_t \sim \mathcal{N}(0_{k \times 1}, I_{k \times k}),$$

and $\mathbf{w}_t \in \mathbb{R}^{p \times 1}$ is the "noise" vector, also sampled from a multivariate normal distribution, i.e.,

$$\mathbf{w}_t \sim \mathcal{N}(0_{p \times 1}, \sigma^2 I_{p \times p}).$$

Furthermore, we assume that all $2n$ random vectors $(\mathbf{z}_t, \mathbf{w}_t, \forall 1 \le t \le n)$ are mutually independent.

In this regime, it is well-known that batch-PCA is asymptotically consistent (hence recovering $A$ up to unitary transformations) with number of samples scaling as $n = O(p)$ Vershynin (2010b). It is interesting to note that in this high-dimensional regime, the signal-to-noise ratio quickly approaches zero, as the signal, or "elongation" of the major axis, $\|Az\|_2$, is $O(1)$, while the noise magnitude, $\|\mathbf{w}\|_2$, scales as $O(\sqrt{p})$. The central goal of this paper is to provide finite sample guarantees for a streaming algorithm that requires memory no more than $O(kp)$ and matches the consistency results of batch PCA in the sampling regime $n = O(p)$ (possibly with additional log factors, or factors depending on $\sigma$ and $k$).

We denote matrices by capital letters (e.g. $A$) and vectors by lower-case bold-face letters ($\mathbf{x}$). $\|\mathbf{x}\|_q$ denotes the $\ell_q$ norm of $\mathbf{x}$; $\|\mathbf{x}\|$ denotes the $\ell_2$ norm of $\mathbf{x}$. $\|A\|$ or $\|A\|_2$ denotes the spectral norm of $A$ while $\|A\|_F$ denotes the Frobenius norm of $A$. Without loss of generality (WLOG), we assume that: $\|A\|_2 = 1$, where $\|A\|_2 = \max_{\|\mathbf{x}\|_2 = 1} \|A\mathbf{x}\|_2$ denotes the spectral norm of $A$. Finally, we write $\langle \mathbf{a}, \mathbf{b} \rangle = \mathbf{a}^\top \mathbf{b}$ for the inner product between $\mathbf{a}, \mathbf{b}$. In proofs the constant $C$ is used loosely and its value may vary from line to line.

**Algorithm 1** Block-Stochastic Power Method     Block-Stochastic Orthogonal Iteration

**input** $\{\mathbf{x}_1, \ldots, \mathbf{x}_n\}$, Block size: $B$

| | |
|---|---|
| 1: $\mathbf{q}_0 \sim \mathcal{N}(0, I_{p \times p})$ (Initialization) | $H^i \sim \mathcal{N}(0, I_{p \times p}), 1 \leq i \leq k$ (Initialization) |
| 2: $\mathbf{q}_0 \leftarrow \mathbf{q}_0 / \|\mathbf{q}_0\|_2$ | $H \leftarrow Q_0 R_0$ (QR-decomposition) |
| 3: **for** $\tau = 0, \ldots, n/B - 1$ **do** | |
| 4: $\quad \mathbf{s}_{\tau+1} \leftarrow 0$ | $S_{\tau+1} \leftarrow 0$ |
| 5: $\quad$ **for** $t = B\tau + 1, \ldots, B(\tau + 1)$ **do** | |
| 6: $\qquad \mathbf{s}_{\tau+1} \leftarrow \mathbf{s}_{\tau+1} + \frac{1}{B}\langle \mathbf{q}_\tau, \mathbf{x}_t \rangle \mathbf{x}_t$ | $S_{\tau+1} \leftarrow S_{\tau+1} + \frac{1}{B}\mathbf{x}_t\mathbf{x}_t^\top Q_\tau$ |
| 7: $\quad$ **end for** | |
| 8: $\quad \mathbf{q}_{\tau+1} \leftarrow \mathbf{s}_{\tau+1} / \|\mathbf{s}_{\tau+1}\|_2$ | $S_{\tau+1} = Q_{\tau+1}R_{\tau+1}$ (QR-decomposition) |
| 9: **end for** | |

**output**

## 4 Algorithm and Guarantees

In this section, we present our proposed algorithm and its finite sample analysis. It is a block-wise stochastic variant of the classical power-method. Stochastic versions of the power method already exist in the literature; see Arora et al. (2012). The main impediment to the analysis of such stochastic algorithms (as in (1)) is the large variance of each step, in the presence of noise. This motivates us to consider a modified stochastic power method algorithm, that has a variance reduction step built in. At a high level, our method updates only once in a "block" and within one block we average out noise to reduce the variance.

Below, we first illustrate the main ideas of our method as well as our sample complexity proof for the simpler rank-1 case. The rank-1 and rank-$k$ algorithms are so similar, that we present them in the same panel. We provide the rank-$k$ analysis in Section 4.2. We note that, while our algorithm describes $\{\mathbf{x}_1, \ldots, \mathbf{x}_n\}$ as "input," we mean this in the streaming sense: the data are no-where stored, and can never be revisited unless the algorithm explicitly stores them.

### 4.1 Rank-One Case

We first consider the rank-1 case for which each sample $\mathbf{x}_t$ is generated using: $\mathbf{x}_t = \mathbf{u}\mathbf{z}_t + \mathbf{w}_t$ where $\mathbf{u} \in \mathbb{R}^p$ is the principal component that we wish to recover. Our algorithm is a block-wise method where all the $n$ samples are divided in $n/B$ blocks (for simplicity we assume that $n/B$ is an integer). In the $(\tau + 1)$-st block, we compute

$$\mathbf{s}_{\tau+1} = \left( \frac{1}{B} \sum_{t=B\tau+1}^{B(\tau+1)} \mathbf{x}_t\mathbf{x}_t^\top \right) \mathbf{q}_\tau. \tag{3}$$

Then, the iterate $\mathbf{q}_\tau$ is updated using $\mathbf{q}_{\tau+1} = \mathbf{s}_{\tau+1} / \|\mathbf{s}_{\tau+1}\|_2$. Note that, $\mathbf{s}_{\tau+1}$ can be computed online, with $O(p)$ operations per step. Furthermore, storage requirement is also linear in $p$.

#### 4.1.1 Analysis

We now present the sample complexity analysis of our proposed method. Using $O(\sigma^4 p \log(p)/\epsilon^2)$ samples, Algorithm 1 obtains a solution $\mathbf{q}_T$ of accuracy $\epsilon$, i.e. $\|\mathbf{q}_T - \mathbf{u}\|_2 \leq \epsilon$.

**Theorem 1.** *Denote the data stream by $\mathbf{x}_1, \ldots, \mathbf{x}_n$, where $\mathbf{x}_t \in \mathbb{R}^p, \forall t$ is generated by (2). Set the total number of iterations $T = \Omega(\frac{\log(p/\epsilon)}{\log((\sigma^2 + .75)/(\sigma^2 + .5))})$ and the block size $B = \Omega(\frac{(1 + 3(\sigma + \sigma^2)\sqrt{p})^2 \log(T)}{\epsilon^2})$. Then, with probability 0.99, $\|\mathbf{q}_T - \mathbf{u}\|_2 \leq \epsilon$, where $\mathbf{q}_T$ is the $T$-th iterate of Algorithm 1. That is, Algorithm 1 obtains an $\epsilon$-accurate solution with number of samples ($n$) given by:*

$$n = \tilde{\Omega}\left( \frac{(1 + 3(\sigma + \sigma^2)\sqrt{p})^2 \log(p/\epsilon)}{\epsilon^2 \log((\sigma^2 + .75)/(\sigma^2 + .5))} \right).$$

Note that in the total sample complexity, we use the notation $\tilde{\Omega}(\cdot)$ to suppress the extra $\log(T)$ factor for clarity of exposition, as $T$ already appears in the expression linearly.

*Proof.* The proof decomposes the current iterate into the component of the current iterate, $\mathbf{q}_\tau$, in the direction of the true principal component (the spike) $\mathbf{u}$, and the perpendicular component, showing that the former eventually dominates. Doing so hinges on three key components: (a) for large enough $B$, the empirical covariance matrix $F_{\tau+1} = \frac{1}{B}\sum_{t=B\tau+1}^{B(\tau+1)} \mathbf{x}_t\mathbf{x}_t^\top$ is close to the true covariance matrix $M = \mathbf{u}\mathbf{u}^\top + \sigma^2 I$, i.e., $\|F_{\tau+1} - M\|_2$ is small. In the process, we obtain "tighter" bounds for $\|\mathbf{u}^\top(F_{\tau+1} - M)\mathbf{u}\|$ for *fixed* $\mathbf{u}$; (b) with probability 0.99 (or any other constant probability), the initial point $\mathbf{q}_0$ has a component of at least $O(1/\sqrt{p})$ magnitude along the true direction $\mathbf{u}$; (c) after $\tau$ iterations, the error in estimation is at most $O(\gamma^\tau)$ where $\gamma < 1$ is a constant.

There are several results that we use repeatedly, which we collect here, and prove individually in the full version of the paper (Mitliagkas et al. (2013)).

**Lemmas 4, 5 and 6**. Let $B$, $T$ and the data stream $\{\mathbf{x}_i\}$ be as defined in the theorem. Then:

- (Lemma 4): With probability $1 - C/T$, for $C$ a universal constant, we have:
$$\left\| \frac{1}{B}\sum_t \mathbf{x}_t\mathbf{x}_t^\top - \mathbf{u}\mathbf{u}^\top - \sigma^2 I \right\|_2 \leq \epsilon.$$

- (Lemma 5): With probability $1 - C/T$, for $C$ a universal constant, we have:
$$\mathbf{u}^\top \mathbf{s}_{\tau+1} \geq \mathbf{u}^\top \mathbf{q}_\tau (1 + \sigma^2)\left(1 - \frac{\epsilon}{4(1+\sigma^2)}\right),$$
where $\mathbf{s}_t = \frac{1}{B}\sum_{B\tau < t \leq B(\tau+1)} \mathbf{x}_t\mathbf{x}_t^\top \mathbf{q}_\tau$.

- (Lemma 6): Let $\mathbf{q}_0$ be the initial guess for $\mathbf{u}$, given by Steps 1 and 2 of Algorithm 1. Then, w.p. 0.99: $|\langle \mathbf{q}_0, \mathbf{u}\rangle| \geq \frac{C_0}{\sqrt{p}}$, where $C_0 > 0$ is a universal constant.

Step (a) is proved in Lemmas 4 and 5, while Lemma 6 provides the required result for the initial vector $\mathbf{q}_0$. Using these lemmas, we next complete the proof of the theorem. We note that both (a) and (b) follow from well-known results; we provide them for completeness.

Let $\mathbf{q}_\tau = \sqrt{1 - \delta_\tau}\mathbf{u} + \sqrt{\delta_\tau}\mathbf{g}_\tau, 1 \leq \tau \leq n/B$, where $\mathbf{g}_\tau$ is the component of $\mathbf{q}_\tau$ that is perpendicular to $\mathbf{u}$ and $\sqrt{1 - \delta_\tau}$ is the magnitude of the component of $\mathbf{q}_\tau$ along $\mathbf{u}$. Note that $\mathbf{g}_\tau$ may well change at each iteration; we only wish to show $\delta_\tau \to 0$.

Now, using Lemma 5, the following holds with probability at least $1 - C/T$:
$$\mathbf{u}^\top \mathbf{s}_{\tau+1} \geq \sqrt{1 - \delta_\tau}(1 + \sigma^2)\left(1 - \frac{\epsilon}{4(1+\sigma^2)}\right). \tag{4}$$

Next, we consider the component of $\mathbf{s}_{\tau+1}$ that is perpendicular to $\mathbf{u}$:
$$\mathbf{g}_{\tau+1}^\top \mathbf{s}_{\tau+1} = \mathbf{g}_{\tau+1}^\top \left( \frac{1}{B}\sum_{t=B\tau+1}^{B(\tau+1)} \mathbf{x}_t\mathbf{x}_t^\top \right) \mathbf{q}_\tau = \mathbf{g}_{\tau+1}^\top (M + E_\tau)\mathbf{q}_\tau,$$

where $M = \mathbf{u}\mathbf{u}^\top + \sigma^2 I$ and $E_\tau$ is the error matrix: $E_\tau = M - \frac{1}{B}\sum_{t=B\tau+1}^{B(\tau+1)} \mathbf{x}_t\mathbf{x}_t^\top$. Using Lemma 4, $\|E_\tau\|_2 \leq \epsilon$ (w.p. $\geq 1 - C/T$). Hence, w.p. $\geq 1 - C/T$:
$$\mathbf{g}_{\tau+1}^\top \mathbf{s}_{\tau+1} = \sigma^2 \mathbf{g}_{\tau+1}^\top \mathbf{q}_\tau + \|\mathbf{g}_{\tau+1}\|_2 \|E_\tau\|_2 \|\mathbf{q}_\tau\|_2 \leq \sigma^2 \sqrt{\delta_\tau} + \epsilon. \tag{5}$$

Now, since $\mathbf{q}_{\tau+1} = \mathbf{s}_{\tau+1}/\|\mathbf{s}_{\tau+1}\|_2$,
$$\begin{aligned}
\delta_{\tau+1} = (\mathbf{g}_{\tau+1}^\top \mathbf{q}_{\tau+1})^2 &= \frac{(\mathbf{g}_{\tau+1}^\top \mathbf{s}_{\tau+1})^2}{(\mathbf{u}^\top \mathbf{s}_{\tau+1})^2 + (\mathbf{g}_{\tau+1}^\top \mathbf{s}_{\tau+1})^2}, \\
&\overset{(i)}{\leq} \frac{(\mathbf{g}_{\tau+1}^\top \mathbf{s}_{\tau+1})^2}{(1 - \delta_\tau)\left(1 + \sigma^2 - \frac{\epsilon}{4}\right)^2 + (\mathbf{g}_{\tau+1}^\top \mathbf{s}_{\tau+1})^2}, \\
&\overset{(ii)}{\leq} \frac{(\sigma^2\sqrt{\delta_\tau} + \epsilon)^2}{(1 - \delta_\tau)\left(1 + \sigma^2 - \frac{\epsilon}{4}\right)^2 + (\sigma^2\sqrt{\delta_\tau} + \epsilon)^2},
\end{aligned} \tag{6}$$

where, $(i)$ follows from (4) and $(ii)$ follows from (5) along with the fact that $\frac{x}{c+x}$ is an increasing function in $x$ for $c, x \geq 0$. Assuming $\sqrt{\delta_\tau} \geq 2\epsilon$ and using (6) and bounding the failure probability with a union bound, we get (w.p. $\geq 1 - \tau \cdot C/T$)

$$\delta_{\tau+1} \leq \frac{\delta_\tau(\sigma^2 + 1/2)^2}{(1 - \delta_\tau)(\sigma^2 + 3/4)^2 + \delta_\tau(\sigma^2 + 1/2)^2} \overset{(i)}{\leq} \frac{\gamma^{2\tau}\delta_0}{1 - (1 - \gamma^{2\tau})\delta_0} \overset{(ii)}{\leq} C_1\gamma^{2\tau}p, \qquad (7)$$

where $\gamma = \frac{\sigma^2 + 1/2}{\sigma^2 + 3/4}$ and $C_1 > 0$ is a global constant. Inequality $(ii)$ follows from Lemma 6; to prove $(i)$, we need the following lemma. It shows that in the recursion given by (7), $\delta_\tau$ decreases at a fast rate. The rate of decrease in $\delta_\tau$ might be initially (for small $\tau$) sub-linear, but for large enough $\tau$ the rate is linear. We defer the proof to the full version of the paper (Mitliagkas et al. (2013)).

**Lemma 2.** *If for any $\tau \geq 0$ and $0 < \gamma < 1$, we have $\delta_{\tau+1} \leq \frac{\gamma^2\delta_\tau}{1 - \delta_\tau + \gamma^2\delta_\tau}$, then,*

$$\delta_{\tau+1} \leq \frac{\gamma^{2t+2}\delta_0}{1 - (1 - \gamma^{2t+2})\delta_0}.$$

Hence, using the above equation after $T = O\left(\log(p/\epsilon)/\log(1/\gamma)\right)$ updates, with probability at least $1 - C$, $\sqrt{\delta_T} \leq 2\epsilon$. The result now follows by noting that $\|\mathbf{u} - \mathbf{q}_T\|_2 \leq 2\sqrt{\delta_T}$. $\qquad \square$

**Remark**: In Theorem 1, the probability of recovery is a constant and does not decay with $p$. One can correct this by either paying a price of $O(\log p)$ in storage, or in sample complexity: for the former, we can run $O(\log p)$ instances of Algorithm 1 in parallel; alternatively, we can run Algorithm 1 $O(\log p)$ times on fresh data each time, using the next block of data to evaluate the old solutions, always keeping the best one. Either approach guarantees a success probability of at least $1 - \frac{1}{p^{O(1)}}$.

## 4.2   General Rank-$k$ Case

In this section, we consider the general rank-$k$ PCA problem where each sample is assumed to be generated using the model of equation (2), where $A \in \mathbb{R}^{p \times k}$ represents the $k$ principal components that need to be recovered. Let $A = U\Lambda V^\top$ be the SVD of $A$ where $U \in \mathbb{R}^{p \times k}$, $\Lambda, V \in \mathbb{R}^{k \times k}$. The matrices $U$ and $V$ are orthogonal, i.e., $U^\top U = I, V^\top V = I$, and $\Sigma$ is a diagonal matrix with diagonal elements $\lambda_1 \geq \lambda_2 \cdots \geq \lambda_k$. The goal is to recover the space spanned by $A$, i.e., $\text{span}(U)$. Without loss of generality, we can assume that $\|A\|_2 = \lambda_1 = 1$.

Similar to the rank-1 problem, our algorithm for the rank-$k$ problem can be viewed as a streaming variant of the classical orthogonal iteration used for SVD. But unlike the rank-1 case, we require a more careful analysis as we need to bound spectral norms of various quantities in intermediate steps and simple, crude analysis can lead to significantly worse bounds. Interestingly, the analysis is entirely different from the standard analysis of the orthogonal iteration as there, the empirical estimate of the covariance matrix is fixed while in our case it varies with each block.

For the general rank-$k$ problem, we use the largest-principal-angle-based distance function between any two given subspaces:

$$\text{dist}\left(\text{span}(U), \text{span}(V)\right) = \text{dist}(U, V) = \|U_\perp^\top V\|_2 = \|V_\perp^\top U\|_2,$$

where $U_\perp$ and $V_\perp$ represent an orthogonal basis of the perpendicular subspace to $\text{span}(U)$ and $\text{span}(V)$, respectively. For the spiked covariance model, it is straightforward to see that this is equivalent to the usual PCA figure-of-merit, the expressed variance.

**Theorem 3.** *Consider a data stream, where $\mathbf{x}_t \in \mathbb{R}^p$ for every $t$ is generated by (2), and the SVD of $A \in \mathbb{R}^{p \times k}$ is given by $A = U\Lambda V^\top$. Let, wlog, $\lambda_1 = 1 \geq \lambda_2 \geq \cdots \geq \lambda_k > 0$. Let,*

$$T = \Omega\left(\log\left(\frac{p}{k\epsilon}\right) \Big/ \log\left(\frac{\sigma^2 + 0.75\lambda_k^2}{\sigma^2 + 0.5\lambda_k^2}\right)\right), \quad B = \Omega\left(\frac{\left(\left(1 + \sigma\right)^2\sqrt{k} + \sigma\sqrt{1 + \sigma^2}k\sqrt{p}\right)^2\log(T)}{\lambda_k^4\epsilon^2}\right).$$

*Then, after $T$ B-size-block-updates, w.p. 0.99, dist$(U, Q_T) \leq \epsilon$. Hence, the sufficient number of samples for $\epsilon$-accurate recovery of all the top-$k$ principal components is:*

$$n = \tilde{\Omega}\left(\frac{\left(\left(1 + \sigma\right)^2\sqrt{k} + \sigma\sqrt{1 + \sigma^2}k\sqrt{p}\right)^2\log(p/k\epsilon)}{\lambda_k^4\epsilon^2\log\left(\frac{\sigma^2 + 0.75\lambda_k^2}{\sigma^2 + 0.5\lambda_k^2}\right)}\right).$$

*Again, we use $\tilde{\Omega}(\cdot)$ to suppress the extra $\log(T)$ factor.*

The key part of the proof requires the following additional lemmas that bound the energy of the current iterate along the desired subspace and its perpendicular space (Lemmas 8 and 9), and Lemma 10, which controls the quality of the initialization.

**Lemmas 8, 9 and 10**. Let the data stream, $A$, $B$, and $T$ be as defined in Theorem 3, $\sigma$ be the variance of noise, $F_{\tau+1} = \frac{1}{B} \sum_{B\tau < t \leq B(\tau+1)} \mathbf{x}_t \mathbf{x}_t^\top$ and $Q_\tau$ be the $\tau$-th iterate of Algorithm 1.

- (**Lemma 8**): $\forall \ \mathbf{v} \in \mathbb{R}^k$ and $\|\mathbf{v}\|_2 = 1$, w.p. $1 - 5C/T$ we have:

$$\|U^\top F_{\tau+1} Q_\tau \mathbf{v}\|_2 \geq (\lambda_k^2 + \sigma^2 - \frac{\lambda_k^2 \epsilon}{4})\sqrt{1 - \|U_\perp^\top Q_\tau\|_2^2}.$$

- (**Lemma 9**): With probability at least $1 - 4C/T$,

$$\|U_\perp^\top F_{\tau+1} Q_\tau\|_2 \leq \sigma^2 \|U_\perp^\top Q_\tau\|_2 + \lambda_k^2 \epsilon/2.$$

- (**Lemma 10**): Let $Q_0 \in \mathbb{R}^{p \times k}$ be sampled uniformly at random as in Algorithm 1. Then, w.p. at least $0.99$: $\sigma_k(U^\top Q_0) \geq C\sqrt{\frac{1}{kp}}$.

We provide the proof of the lemmas and theorem in the full version (Mitliagkas et al. (2013)).

## 4.3 Perturbation-tolerant Subspace Recovery

While our results thus far assume $A$ has rank exactly $k$, and $k$ is known *a priori*, here we show that both these can be relaxed; hence our results hold in a quite broad setting.

Let $\mathbf{x}_t = A\mathbf{z}_t + \mathbf{w}_t$ be the $t$-th step sample, with $A = U\Lambda V^T \in \mathbb{R}^{p \times r}$ and $U \in \mathbb{R}^{p \times r}$, where $r \geq k$ is the unknown true rank of $A$. We run Algorithm 1 with rank $k$ to recover a subspace $Q_T$ that is contained in $U$. The largest principal angle-based distance, from the previous section, can be used directly in our more general setting. That is, $\text{dist}(U, Q_T) = \|U_\perp^T Q_T\|_2$ measures the component of $Q_T$ "outside" the subspace $U$.

Now, our analysis can be easily modified to handle this case. Naturally, now the number of samples we require increases according to $r$. In particular, if

$$n = \tilde{\Omega}\left( \frac{\left((1+\sigma)^2\sqrt{r} + \sigma\sqrt{1+\sigma^2}r\sqrt{p}\right)^2 \log(p/r\epsilon)}{\lambda_r^4 \epsilon^2 \log\left(\frac{\sigma^2 + 0.75\lambda_r^2}{\sigma^2 + 0.5\lambda_r^2}\right)} \right),$$

then $\text{dist}(U, Q_T) \leq \epsilon$. Furthermore, if we assume $r \geq C \cdot k$ (or a large enough constant $C > 0$) then the initialization step provides us better distance, i.e., $\text{dist}(U, Q_0) \leq C'/\sqrt{p}$ rather than $\text{dist}(U, Q_0) \leq C'/\sqrt{kp}$ bound if $r = k$. This initialization step enables us to give tighter sample complexity as the $r\sqrt{p}$ in the numerator above can be replaced by $\sqrt{rp}$.

## 5 Experiments

In this section, we show that, as predicted by our theoretical results, our algorithm performs close to the optimal batch SVD. We provide the results from simulating the spiked covariance model, and demonstrate the phase-transition in the probability of successful recovery that is inherent to the statistical problem. Then we stray from the analyzed model and performance metric and test our algorithm on real world–and some very big–datasets, using the metric of explained variance.

In the experiments for Figures 1 (a)-(b), we draw data from the generative model of (2). Our results are averaged over at least 200 independent runs. Algorithm 1 uses the block size prescribed in Theorem 3, with the empirically tuned constant of $0.2$. As expected, our algorithm exhibits linear scaling with respect to the ambient dimension $p$ – the same as the batch SVD. The missing point on batch SVD's curve (Figure 1(a)), corresponds to $p > 2.4 \cdot 10^4$. Performing SVD on a dense $p \times p$ matrix, either fails or takes a very long time on most modern desktop computers; in contrast, our streaming algorithm easily runs on this size problem. The phase transition plot in Figure 1(b)

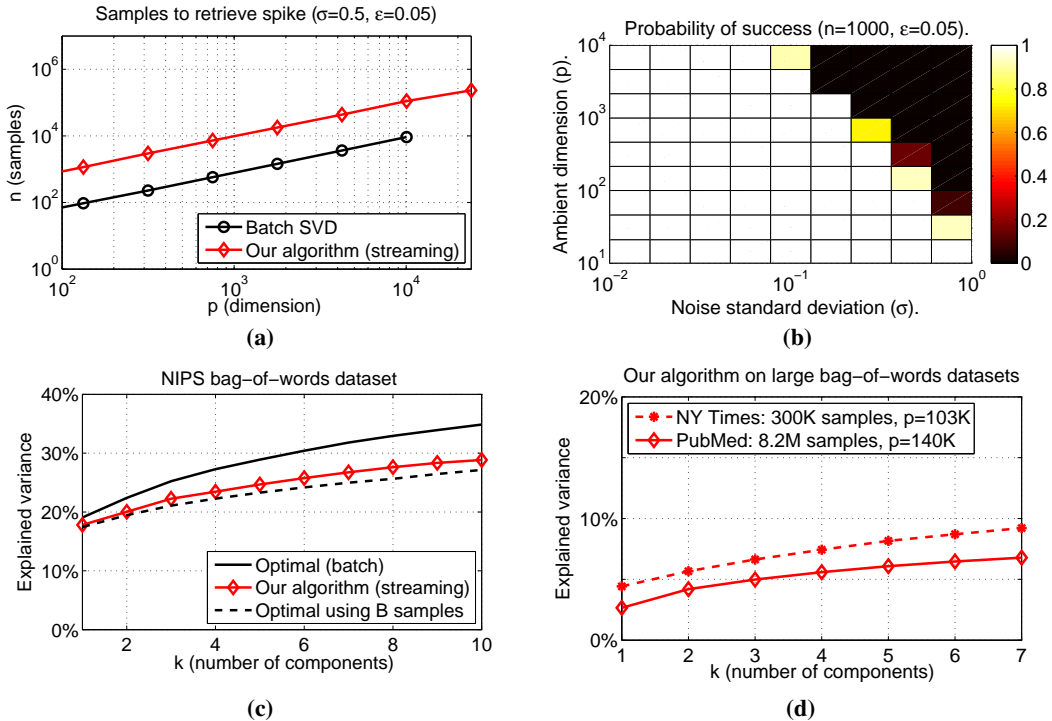

Figure 1: **(a)** Number of samples required for recovery of a single component ($k = 1$) from the spiked covariance model, with noise standard deviation $\sigma = 0.5$ and desired accuracy $\epsilon = 0.05$. **(b)** Fraction of trials in which Algorithm 1 successfully recovers the principal component ($k = 1$) in the same model, with $\epsilon = 0.05$ and $n = 1000$ samples, **(c)** Explained variance by Algorithm 1 compared to the optimal batch SVD, on the NIPS bag-of-words dataset. **(d)** Explained variance by Algorithm 1 on the NY Times and PubMed datasets.

shows the empirical sample complexity on a large class of problems and corroborates the scaling with respect to the noise variance we obtain theoretically.

Figures 1 (c)-(d) complement our complete treatment of the spiked covariance model, with some out-of-model experiments. We used three bag-of-words datasets from Porteous et al. (2008). We evaluated our algorithm's performance with respect to the fraction of explained variance metric: given the $p \times k$ matrix $V$ output from the algorithm, and all the provided samples in matrix $X$, the fraction of explained variance is defined as $\mathrm{Tr}(V^T X X^T V) / \mathrm{Tr}(X X^T)$. To be consistent with our theory, for a dataset of $n$ samples of dimension $p$, we set the number of blocks to be $T = \lceil \log(p) \rceil$ and the size of blocks to $B = \lfloor n/T \rfloor$ in our algorithm. The NIPS dataset is the smallest, with $1500$ documents and 12K words and allowed us to compare our algorithm with the optimal, batch SVD. We had the two algorithms work on the document space ($p = 1500$) and report the results in Figure 1(c). The dashed line represents the optimal using $B$ samples. The figure is consistent with our theoretical result: our algorithm performs as well as the batch, with an added $\log(p)$ factor in the sample complexity.

Finally, in Figure 1 (d), we show our algorithm's ability to tackle very large problems. Both the NY Times and PubMed datasets are of prohibitive size for traditional batch methods – the latter including $8.2$ million documents on a vocabulary of $141$ thousand words – so we just report the performance of Algorithm 1. It was able to extract the top 7 components for each dataset in a few hours on a desktop computer. A second pass was made on the data to evaluate the results, and we saw 7-10 percent of the variance explained on spaces with $p > 10^4$.

## Footnotes

[1]This is similar to what is sometimes referred to as the *single pass* model.

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
