[Reviews · NeurIPS 2013]

Submitted by Assigned_Reviewer_1

Summary: proposes an approach to one-pass SVD based on a blocked variant of the power method, which variance is reduced within each block of streaming data, and compares to exact batch SVD.
Figure 1d is offered as an example where the proposed Algo 1 can scale to data for which the authors claim to be so large that "traditional batch methods" could not be run and reported. Yet there are many existing well-known SVD methods which are routinely used for even larger data sets than the largest here (sparse 8.2M vectors for 120k dimensions). These include the EMPCA (Roweiss 1998) and fast randomized SVD (Haiko et at 2011), both of which the author's cite. Why were these methods (both very simple to implement efficiently even in Matlab, etc.) not reported for this data? Especially necessary to compare against is the randomized SVD, since it too can be done in one-pass (see Haiko et al); although that cited paper discusses the tradeoffs in doing multiple passes -- something this paper does not even discuss. The authors say it took "a few hours" for Algo 1 to extract the top 7 components. Methods like the randomized SVD family of Haiko et al scale linearly in those parameters (n=8.2M and d=120k and k=7 and the number of non-zeros of the sparse data) and typically run in less than 1 hour for even larger data sets. So, demonstrating both the speed and accuracy of the proposed Algo 1 compared to the randomized algorithms seems necessary at this point, to establish the practical significance of this proposed approach.

Even though the paper argues that previous approaches do not have "finite sample guarantees", this does not excuse them from having to demonstrate that their approach is actually better in practice than existing methods. As written, this paper basically presents yet another way to do SVD, for which there are already many existing methods, including streaming one-pass methods (such as the above mentioned randomized SVD).

Also, the focus on spiked covariance models in the theory parts of this paper is confusing. What is the point of proving finite sample complexity results for that case, but then failing to compare to existing state of the art methods (like EMPCA and randomized SVD) for data (like PubMed) which does not fall into the spiked covariance type of data?
Summary: A paper addresses an important, timely problem (fast one-pass PCA), but fails to compare to existing methods for that, making it hard to judge the significance of this result. Also, a lot of theory is offered, but their significance is also not clear, in part because of its focus on a spiked covariance model who's relevance to real data is not explained.

Submitted by Assigned_Reviewer_5

Summary: This paper identifies and resolves a basic gap in the design of streaming PCA algorithms. It is shown that a block stochastic streaming version of the power method recovers the dominant rank-k PCA subspace with optimal memory requirements and sample complexity not too worse than batch PCA (which maintains the covariance matrix explicitly), assuming that streaming data is drawn from a natural probabilistic generative model. The paper is excellently written and provides intuitions for the analysis, starting with exact rank 1 and exact rank k case to the general rank k approximation problem. Some empirical analysis is also provided illustrating the approach for PCA on large document-term matrices.

Questions/Comments:

- Will the algorithm tolerate approximate orthogonalization (QR step in Algorithm 1) which now becomes the main bottleneck for large p and moderate sized k.

- The closest related algorithmic line of work is on sketching and sampling schemes for generating low-rank approximations to streaming matrices (e.g., Clarkson and Woodruff). Please elaborate more on the "fundamental sample complexity reasons" for why these approaches are algorithmically weaker, or whether their guarantees may be strengthened for the spiked covariance model.

- Are there generative models other than the spiked covariance model for which the batch PCA sample complexity has been characterized? Please comment on whether the proof techniques used in the paper can be generalized to other models.

- I am a bit surprised that generating rank=7 approximations for the NYTimes and Pubmed problems takes several hours. What is the computational bottleneck in practice?

- The gap between optimal model and the streaming algorithm in Figure (c) begs the question of whether results can be improved in a semi-streaming setting, where data is stored on disk and you are allowed to make a small number of bounded memory passes.

Minor typos:

Remark above section 4.2: "runO(logp)" -> run O(log p)
Summary: Very well written paper on streaming PCA algorithms with optimal memory requirements and sample complexity similar to batch PCA, closing a gap in the literature. The proof techniques would likely find use elsewhere, and the algorithms are practical.

Submitted by Assigned_Reviewer_6

The authors present an algorithm for PCA in the streaming model, under the spiked covariance model (essentially, a low-rank matrix generating the samples that PCA seeks to recover). The proposed algorithm has optimal storage guarantees (hence the streaming model setting), and comes with strong sample complexity guarantees (essentially linear in the dimensionality of the data points).

The theory in the paper is solid and the authors have done a good job of presenting their results within the context of existing state-of-the-art. The theory is clearly the strong suit of the paper; the experimental evaluation and the comparisons to standard batch PCA approaches are useful, but not sufficient. I would prefer to see a lot more details on the performance of the method on the large datasets (e.g., the authors could use an approximate method to compute PCA and compare to the output of their algorithm, instead of simply reporting the percentage of explained variance). The full version of the paper (submitted as supplementary material) did not provide much additional experimental details either.
Summary: A very good theoretical paper on PCA in the streaming model under the spiked covariance model. The theoretical guarantees are strong, but the experimental evaluation is lacking.
Author Feedback

Author rebuttal: Response to reviewer 1:
-----------
Regarding the comparison to other algorithms -- including but not limited to EMPCA and the randomized SVD by Halko et al.:

- We first want to emphasize that our main focus and contribution has been the study of PCA in a theoretically rigorous manner in the high-dimensional regime and with limited memory. For such high-d, streaming PCA problems, understanding has been extremely lacking, so we started this study by using the spiked covariance model, which is strong and popular, yet simple. Existing methods like that of Halko et al. are designed for worst case matrices. Their guarantees are weak in the high-d setting, where signal to noise ratio is very poor. In fact, as explained below, their method reduces to the normal batch SVD when we require exact recovery. We aim for something better than the worst case; in our experience, real data is not nearly as bad as that.

Comparison to randomized SVD by Halko et al.: Halko et al's error bound includes a term like (\sum_{j=k+1} sigma_j^2)/B, where "B" is the oversampling parameter and sigma_j's are the singular values of the data matrix. For the spiked covariance matrix this implies that the number of samples they need *in memory* is O((p-k)sigma^2) where p is the dimensionality and sigma is the variance of the Gaussian noise. Hence, for a constant noise, their method requires O(p) samples in memory, that is a *O(p^2) memory requirement* -- prohibitive in high-d.

Aside from the bounds, it is also intuitive that sketching-based methods are not suited for a statistical setting. With samples from a stochastic model, subsampling and/or randomized dimensionality reduction (a la Halko et al., Clarkson and Woodruff, etc.) amounts to taking fewer samples. For example, randomized SVD computes A'=A \Omega where \Omega is a zero mean matrix. Note that if \Omega is a +1/-1 matrix then A' and A are statistically *indistinguishable*; no improvement is expected by randomization.

Comparison to EMPCA: As mentioned in the paper, EMPCA eludes theoretical analysis. As with many other methods, its hardness lies in bounding the variance at each step. Our method handles this issue by doing block updates.

Regarding our focus on the spiked covariance model:

- The spiked covariance model is a reasonable description of how things look in the presence of a small number of signal directions and ambient subgaussian noise. Any kind of spectrum can be treated as an instance of the spiked covariance model, by regarding the magnitude of the first dropped component (the k+1-th component) as the noise magnitude. This will give a valid bound on the samples required, and in some cases, mean that the model is, indeed, even pessimistic.

In Figure 1(c), we compare the performance of our algorithm to the optimal, on real data, and consider the explained variance. Those experiments suggest that, even outside the analyzed cases, our algorithm does well, again exhibiting our theoretically proven log(p) gap from the optimal.



Response to reviewer 2:
-----------
Our answers to her/his questions in the order they were provided:
"Will the algorithm tolerate approximate orthogonalization [...]?"

- We thank the reviewer for this insight. Our proof can be easily modified to include error in the QR approximation; in particular, a \delta error in QR leads to a roughly (\delta \log(1/\epsilon)) additional error in our analysis.

"The closest related algorithmic line of work is on sketching and sampling schemes for generating low-rank approximations to streaming matrices (e.g., Clarkson and Woodruff). Please elaborate more on the 'fundamental sample complexity reasons' for why these approaches are algorithmically weaker, or whether their guarantees may be strengthened for the spiked covariance model."

- Subsampling methods, i.e. dropping samples, are fundamentally suboptimal when the objective is minimizing the sample complexity. Regarding Clarkson and Woodruff specifically, their results are expressed in the Frobenius norm. In the high-d setting this leads to very poor bounds. For the spiked covariance model, their method leads to an error bound of O(p) for component retrieval; we provide constant bounds.


"Are there generative models other than the spiked covariance model for which the batch PCA sample complexity has been characterized? [...]"

- We have not seen any such results beyond the spiked covariance model for the batch case; our understanding is that this happens because the spiked covariance model is capable of encompassing many important cases. Our proofs can be used to provide bounds for models with general population spectrum, subgaussian mixing vectors (z_t in our model) and subgaussian noise.


"[...] What is the computational bottleneck in practice?"

- Our proof-of-concept implementation is not optimized for computational performance and reading the data from the disk was the bottleneck. The PubMed dataset is almost 8GB big -- twice as big as our system's RAM; the entries had to be read from the disk line-by-line and we used simple but suboptimal system calls through MATLAB for that purpose.


"[...] whether results can be improved in a semi-streaming setting, where data is stored on disk and you are allowed to make a small number of bounded memory passes."

- Definitely. Assuming you are allowed to make $l$ passes, the total sample complexity would be $l$ times smaller. In the case when $l=O(\log(p))$, we revert back to the classic power method.



Response to reviewer 3:
-----------
We will elaborate on the performance on very large datasets in future versions, trying the reviewer's suggestion of an approximate method as the baseline. We would again stress that, so far, the focus has been to show that, where the streaming high-d setting had eluded analysis, a simple block-update-based method can lead to an effective practical solution, along with strong guarantees.